# Simultaneous Qualitative and Quantitative Evaluation of the Coptidis Rhizoma and Euodiae Fructus Herbal Pair by Using UHPLC-ESI-QTOF-MS and UHPLC-DAD

**DOI:** 10.3390/molecules25204782

**Published:** 2020-10-18

**Authors:** Yan-Ying Li, Min-Qun Guo, Xue-Mei Li, Xiu-Wei Yang

**Affiliations:** 1State Key Laboratory of Natural and Biomimetic Drugs (Peking University), Department of Natural Medicines, School of Pharmaceutical Sciences, Peking University Health Science Center, Peking University, Beijing 100191, China; yanyingli@ykrskj.com; 2TCM R&D Center, Beijing Increase Pharm Co. Ltd., Beijing 102200, China; guominqun@ykrskj.com; 3TCM R&D Center, Beijing Increase Innovative Drug Co. Ltd., Beijing 102200, China

**Keywords:** Coptidis Rhizoma, Euodiae Fructus, herbal pair, quality markers, UHPLC-ESI-QTOF-MS, UHPLC-DAD fingerprint

## Abstract

The herbal pair of Coptidis Rhizoma (CR) and Euodiae Fructus (EF) is a classical traditional Chinese medicine formula used for treating gastro-intestinal disorders. In this study, we established a systematic method for chemical profiling and quantification analysis of the major constituents in the CR-EF herbal pair. A method of ultra high performance liquid chromatography/quadrupole time-of-flight mass spectrometry (UHPLC-QTOF-MS) for qualitative analysis was developed. Sixty-five compounds, including alkaloids, phenolics, and limonoids, were identified or tentatively assigned by comparison with reference standards or literature data. The UHPLC fingerprints of 19 batches of the CR-EF herbal pair samples were obtained and the reference fingerprint chromatograms were established. Furthermore, nine compounds among 24 common peaks of fingerprints were considered as marker components, which either had high contents or significant bioactivities, were applied to quality control of the CR-EF herbal pair by quantitative analysis. This UHPLC-DAD analysis method was validated by precision, linearity, repeatability, stability, recovery, and so on. The method was simple and sensitive, and thus reliable for quantitative and chemical fingerprint analysis for the quality evaluation and control of the CR-EF herbal pair and related traditional Chinese medicines.

## 1. Introduction

Herbal medicines have been playing an important role in human health since ancient times, and its therapeutic theory and clinical effects are contained in precious traditional medical knowledge reserves [1]. Among them Traditional Chinese Medicine (TCM) occupies a significant position in global healthcare systems, and robust quality assessment and control over its complex chemical composition are of great importance to assure its efficacy and safety. Over the past two decades, the aim of TCM chemical fingerprints to obtain a wholistic characterization of complex chemical matrices has become one of the most convincing tools for the quality control and assessment of TCM [1].

Coptidis Rhizoma (CR, the rhizomes of *Coptis chinensis* Franch.) and Euodiae Fructus (EF, the nearly ripe fruits of *Euodia rutaecarpa* Benth.), as a classic TCM, were often used together in what is called the “CR-EF couple”, and could compose different functional TCM formulae by different ratios. When the proportion is 6:1 (CR-EF, *w/w*), it is called Zuojin formula. Zuojin Pill, or one of Zuojin formula products, and has been used in clinical treatment of gastrointestinal disorders, such as gastric ulcer, gastroesophageal reflux disease, gastritis, pyloric obstruction, etc., which was first recorded in “Danxi’s experiential therapy” in the 15th century and still widely used in China today [2,3,4].

The results of pharmacological research showed that the bioactivities of Zuojin pill are attributed to alkaloids from the herbal pair of CR and EF [5]. The active ingredients of CR are protoberberine-type alkaloids and those of EF are indolequinoline and quinolone alkaloids. Further investigation has revealed the presence of 22 alkaloids [2,6]: (1) berberastine, (2) epiberberine, (3) jatrorrhizine, (4) columbamine, (5) palmatine, (6) coptisine, (7) berberine, (8) coptichic aldehyde, (9) coptichine, (10) 13-carboxaldehyde-8-oxocoptisine, and (11) 8-oxocoptisine, which are from CR; (12) dehydroevodiamine, (13) evodiamine, (14) rutaecarpine, (15) 1-methyl-2-nonyl-4(1*H*)-quinolone, (16) 1-methyl-2-[(*Z*)-6-undecenyl]-4(1*H*)-quinolone, (17) 1-methyl-2-undecyl-4(1*H*)-quinolone, (18) evocarpine, (19) 1-methyl-2-[(6*Z*,9*Z*)-6,9-pentadecadienyl]-4(1*H*)-quinolone, (20) dihydroevocarpine, (21) 1-methyl-2-[(*Z*)-10-pentadecenyl]-4(1*H*)-quinolone, and (22) 1-methyl-2-[(*Z*)-6-pentadecenyl]-4(1*H*)- quinolone, which are from EF.

It is reported that the quantities of seven alkaloids in ethanol extract [4] of Zuojin pill are epiberberine 32.65 ± 0.43 mg·g^−1^, jatrorrhizine 13.59 ± 0.15 mg·g^−1^, coptisine 110.64 ± 1.15 mg·g^−1^, palmatine 61.20 ± 0.79 mg·g^−1^, berberine 153.0 ± 1.94 mg·g^−1^, evodiamine 1.89 ± 0.026 mg·g^−1^, and rutaecarpine 1.47 ± 0.050 mg·g^−1^, while the quantities of six alkaloids in the water extract [7] of Zuojin pill are columbamine 2.81 ± 0.08 mg·g^−1^, epiberberine 3.90 ± 0.07 mg·g^−1^, jatrorrhizine 2.70 ± 0.04 mg·g^−1^, coptisine 6.85 ± 0.08 mg·g^−1^, palmatine 11.02 ± 0.11 mg·g^−1^, and berberine 35.86 ± 0.72 mg·g^−1^.

Although alkaloids are major ingredients of both CR and EF, the quality marks of them are still unclear. Until now there has been no report concerning the quality control of the CR-EF herbal pair. Therefore, it is critically important to establish an effective analytical method which enables the comprehensive quality assessment of Zuojin formula for ensure its safety, efficacy, and batch-to-batch consistency.

In this study, we established an UHPLC-QTOF-MS method for rapidly separating and reliably identifying the components in the CR-EF herbal pair for the first time. A total of 65 compounds in the CR-EF herbal pair were identified or deduced based on their retention times and MS spectra data. Based on the qualitative analysis results of CR-EF herbal pair, a new method for simultaneously fingerprinting and quantitative analysis has been developed by UHPLC-DAD. Nine representative compounds, including dehydroevodiamine, coptisine, epiberberine, columbamine, jatrorrhizine, berberine, palmatine, evodiamine, and rutaecarpine, which either had high contents or significant bioactivities, out of 24 common peaks of fingerprint were applied as marker compounds to assess the quality of the CR-EF herbal pair. This validated method was further applied to determine 19 batches of CR-EF herbal pair samples. It is expected that this research will provide comprehensive analysis information for the quality control of CR-EF herbal pair products and its crude drugs and, furthermore, provide a basis for guiding new drug development of effective CR-EF herbal pair compositions.

## 2. Results and Discussion

### 2.1. Selection of Extraction and Analytical Conditions

In order to obtain efficient extraction of the compounds, variable parameter involved in this procedure such as extraction method (ultrasonication, reflux), extraction solvent (chloroform, methanol, and a serial concentration of methanol-hydrochloric acid), solvent volume (50, 100, and 120 mL), and soaking and extraction time (30, 45, and 60 min, respectively) were performed. The final selected method was to soak 0.7 g of the CR-EF herbal pair powder with 100 mL of MeOH-HCl solution (100:1; *v/v*) for 1 h, and then ultrasonicated at room temperature for 30 min.

The UHPLC conditions were tried by using various type of column, column temperature, mobile phase system and flow rate. After comparing different brands of columns, including GL Science ODS-3, Ultimate UHPLC AQ-C18, and Agilent ZORBAX-SB-C18, the Ultimate UHPLC AQ-C18 column finally satisfied the separation of characteristic peaks. In addition, different kinds of mobile phases, such as acetonitrile and water with various modifiers (including formic acid, phosphoric acid, and trifluoroacetic acid (TFA)) were tested. It was found that the suitable mobile phase is the mixture of acetonitrile and 0.01% (*v/v*) TFA aqueous solution, which made it possible to separate 24 compounds simultaneously. Considering TFA was unsuitable for the MS system, 0.05% formic acid was selected for MS analysis. Five detection wavelengths (230, 254, 275, 300, 330 nm) were selected according to the maximum absorptions of quantified components in the UV spectra from a DAD coupled to the UHPLC system. For the MS conditions, capillary voltage, fragmentor voltage, and collision energy were optimized, and both positive and negative ion modes were employed to identify the corresponding signals.

### 2.2. UHPLC-QTOF-MS Qualitative Analysis of CR-EF Herbal Pair

To characterize the chemical constituents in CR-EF herbal pair extracts, an UHPLC-ESI-QTOF-MS method was established. As can be seen in Figure 1 and Appendix A, a total of 65 compounds were identified or tentatively characterized, 10 of them were identified unambiguously by comparing their retention times and accurate masses with data from the corresponding reference standards. The structures of the other 55 compounds were tentatively characterized by comparing their characteristic high resolution mass data with the data from previous publications. All these compounds were divided into 3 types based on the structural characteristics, including 44 alkaloids, seven limonoids, and 14 phenolic compounds. The mass error for molecular ions of all identified compounds (except compound **43**) was within ± 5 ppm, the total ion chromatograms (TIC) in negative and positive ion modes were displayed in Figure 1. 

#### 2.2.1. Identification of Alkaloids

In the present study, a total of 44 alkaloids are mainly assigned to three classes of isoquinoline, indoloquinazoline and quinolone according to the deduced structures by the positive ion mode of QTOF-MS.

##### Identification of Isoquinoline Alkaloids in CR

Twenty-two compounds were identified based on the fragmentation pathway from the previous reports (Figure 2) [8,9]. Compounds **3**, **20**, **25**, **26**, **27**, **30**, and **31** were identified to be magnoflorine, coptisine, epiberberine, columbamine, jatrorrhizine, berberine, and palmatine by comparing with the reference standards. To successfully identify and confirm the other alkaloids, the MS fragmenting patterns of above compounds were investigated.

As for the protoberberine alkaloids as the basic structure, if position 3 is -OCH_3_, such as jatrorrhizine (**27**), -CH_3_ at the terminal position will be lost, and position 2 and 3 both have -OCH_3_ groups, and often lost CH_4_ to form a 1,3-dioxypentacyclic ring, such as jatrorrhizine (**27**) and palmatine (**31**). If position 9 is -OCH_3_, such as components jatrorrhizine (**27**), palmatine (**31**), and berberine (**30**), the -CH_3_ at their terminal positions will fall off. In addition, the benzene ring A beside the 1,3-dioxo five-membered ring is cracked, losing the -CO fragment, and continuously losing -CH_2_ fragments, finally forming 277/278 fragment ions [8,9].

Taking berberine as an example, the molecular formula C_20_H_18_NO_4_^+^ shows a positive molecular ion at 336 [M]^+^ with relative small fragment abundance, which loses the terminal methyl group, and hydrogen rearrangement produces fragment ion 320 [M – CH_4_]^+^ with a high abundance ratio. On the one hand, the fragment can lose the methyl group of the terminus, forming the 306 [M − CH_4_ − CH_2_]^+^ fragment ion. On the other hand, one side of the benzene ring can also be cracked and lose -CO fragments, forming the 292 [M − CH_4_ − CO]^+^ fragment ion, and continuing to lose -CH_2_ and -CH_3_ fragments, forming [M − CH_4_ – CO − CH_2_]^+^ fragment ion [10]. Consequently, by matching the accurate masses and the fragment ions with those of previous studies [11,12,13,14], compounds **11**, **12**, **16**, **17**, **20**, **29**, **32**, and **39** were tentatively assigned as 9,10-dihydroxycolumbamine, berberastine, berberrubine, 2-hydroxyjatrorrhizine, demethylene coptisine, groenlandicine, 13-methylberberine, and 8-methoxyberberine phenolbetaine, respectively.

The [M + H]^+^ ion of compound **34**, which was observed at *m*/*z* 542.1812 (C_31_H_27_NO_8_), was identical to those of coptichine, which has been previously isolated from CR-EF herbal pair extract [2]. 

Compounds **23**, **24**, and **35** have the same molecular weight and fragmentation pattern, and similar LC retention behavior to those of 2-hydroxy jatrorrhizine, groenlandicine, and coptichine. Thus, these were tentatively assigned as isomers of 2-hydroxy jatrorrhizine, groenlandicine and coptichine.

Compound **44** generated an [M + H]^+^ ion at 352.1184 (C_20_H_17_NO_5_), and the fragments of this compound were identical to those of 8-oxoberberine. So, compound **44** was identified as 8-oxoberberine [15]. Compound **15** has the identical molecular weight and fragmentation pattern of compound **44**, so this was tentatively assigned as an isomer of 8-oxoberberine, 8-oxoepiberberine. 

Compound **13**, according to the exact molecular weight and the chemical formula, we could deduce that it had a protoberberine type skeleton. The fragment ion at *m/z* 338.1013 by loss of CH_4_ (−16 Da) from the protonated molecular ion further confirmed the presence of a terminal methyl group. In addition, the appearance of fragment ions *m/z* 324.0858 [M + H − CH_4_ − CH_2_]^+^, 296.0881 [M + H − CH_4_ − CH_2_ − CO]^+^and 278.0747 [M + H − CH_4_ − CH_2_ – CO − CH_4_]^+^ indicated the presence of vicinal methoxy groups at the A or D-ring and the fragment ion 310.1070 [M + H − CH_4_ − CO]^+^ indicated the presence of vicinal methoxy and hydroxy groups at the D or A-ring, respectively. Based on these data, Compound **13** was tentatively identified as sinotumine H, which was reported from CR for the first time. Accordingly, the structure of sinotumine H was defined as 9-demethyloxypalmatine **[16]**. 

##### Identification of Quinolone Alkaloids in EF

According to the fragmentation pathway and information, 15 compounds were identified (Appendix A). Taking compound **51** as an example, it was exhibited [M + H]^+^ ion at *m/z* 284.2007 in the positive ion mode, indicating an MW of 283. In the MS/MS spectrum, a series of characteristic ions *m/z* 159.0671 ([C_10_H_9_NO]^+^), 173.0828 ([C_10_H_9_NO + CH_2_]^+^) and 186.0906 ([C_10_H_9_NO + C_2_H_3_]^+^) were yielded, respectively, by fragmentation of different positions at side chain, as shown in Figure 3. In addition, the fragment ion at *m/z* 200.1077 was generated by neutral loss of C_6_H_12_ from the parent nucleus. Therefore, compound **51** was tentatively identified as 1-methyl-2-[(*Z*)-4-nonenyl]-4(1*H*)-quinolone, and previously reported from this plant [17].

Similarly, compounds **53**–**65** shared the same fragmentation behavior as compound **51**, and were thus tentatively identified as 1-methyl-2-nonyl-4(1*H*)-quinolone, 1-methyl-2-[(*Z*)-6-undecenyl]-4(1*H*)-quinolone, 1-methyl-2-[(4*Z*,7*Z*)- tridecadienyl]-4(1*H*)-quinolone, 1-methyl-2-[(*E*)-1-undecenyl]-4(1*H*)-quinolone, 1-methyl-2-undecyl-4(1*H*)-quinolone, 1-methyl-2-[(6*Z*,9*Z*,12*E*)-pentadeca triene]-4(1*H*)-quinolone, evocarpine, 1-methyl-2-[(6*Z*,9*Z*)-pentadecadienyl]-4(1*H*)-quinolone, 1-methyl-2-[13-tetradecenyl]-4-(1*H*)-quinolone, dihydroevocarpine, 1-methyl-2-[(*Z*)-9-pentadecenyl]-4(1*H*)-quinolone, 1-methyl-2-tetradecy-4-(1*H*)-quinolone, 1-methyl-2-pentadecyl-4(1*H*)-quinolone [18]. Additionally, for compound **52**, it displayed a [M + H]^+^ ion at *m/z* 358.2739 (C_23_H_35_NO_2_). In the MS/MS experiment, except the above quinolone skeleton specific fragment ions, compound **52** generated a [M + H − H_2_O]^+^ fragment at *m/z* 340.2624 by losing 18 Da because the -OH moiety was easy to separate from the precursor ion. Thus, it was identified as 1-methyl-2-[12-hydroxy-tridecyl]-4(1*H*)-quinolone comparing with the literature [17].

##### Identification of Indoloquinazoline in EF

Seven compounds were identified based on the fragmentation pathway (Figure 4), of which compounds **18**, **46**, and **48**, were unambiguously identified as dehydroevodiamine, evodiamine and rutaecarpine by comparing their retention time and MS data with those of reference standards. Other compounds were tentatively assigned on the basis of their empirical molecular formulae and literature report. Take compound **42** as an example, it displayed an [M + H]^+^ ion at *m/z* 308.1761 (C_19_H_22_N_3_O). In the MS/MS experiment, the obtained ion produced fragments at *m/z* 134.0603 [C_8_H_8_NO]^+^, 116.0500 [C_8_H_6_N]^+^ and 106.0655 [C_7_H_8_N]^+^. Thus, it was identified as evodiamide by comparing the ion fragments with the literature [17]. Compound **41** exhibited an [M + H]^+^ ion at *m/z* 294.1602 (C_18_H_19_N_3_O), which was 14 Da less than that of compound **42**, and also produced the same specific three ion fragments as that of compound **42**. On the basis of the available reference data [18], compound **41** was tentatively identified as *N*^b^-demethyl evodiamide. Similarly, based on the exact mass and characteristic fragment ions of compound **50**, tentatively identified as goshuyuamide I. Compound **49**, it displayed an [M + H]^+^ ion at *m/z* 318.1239 (C_19_H_15_N_3_O_2_), which was 30 Da more than that of compound **48**, and generated an intensive fragment at 286.0981 [M + H − CH_3_OH]^+^, suggesting compound **49** has a terminal methoxyl group. It was tentatively deduced as hortiacine based on the exact mass, molecular formulae, and literature report [19].

#### 2.2.2. Identification of Limonoids

Limonoids are another major group of chemical constituents from EF. The MS and MS/MS spectra of EF obtained by QTOF-MS, and its positive ion mode showed higher sensitivity compared with the negative ion mode, allowing the identification of seven compounds. Among them, compound **38** was identified to be limonin by comparison with the reference standard, which generated limonin type characteristic fragments (see Figure 5) at *m/z* 161.0 [C_9_H_5_O_3_]^+^, 114.03 [C_5_H_6_O_3_]^+^ and 95.0 [C_5_H_3_O_2_]^+^.

Compounds **36**, **37**, **40**, **43**, and **47** all displayed the characteristic fragments 161.0 and 95.0, this indicated that they are limonin type limonoids, and were also detected or isolated from EF extract. Their MS/MS fragments are similar to the reported literature. Thus, they are tentatively assigned as rutaevine, 12α-hydroxylimonin, evodol, glaucin A, and 6-acetoxy-5-epilimonin, which were also isolated from the fruit of *Evodia rutaecarpa* Bentham [18,19,20,21].

Compound **45** gave an [M − H]^−^ ion at *m/z* 515.1918 corresponding to the chemical formula of C_27_H_32_O_10_. The appearance of five main MS/MS fragment ions at *m/z* 515.1934 [M − H]^−^, 427.1409 [M – H − 2CO_2_]^−^, 401.0887 [M – H − C_5_H_6_O_3_]^−^, 357.0601 [M – H − C_5_H_6_O_3_ − CO_2_]^−^, 313.0738 [M – H − C_5_H_6_O_3_ − 2CO_2_]^−^ and 269.0830 [M – H − C_5_H_6_O_3_ − 2CO_2_ − CH_4_ − CO]^−^ were yielded. Thus, compound **45** tentatively characterized as evorubodinin, which has been previously isolated from EF extract [22].

#### 2.2.3. Identification of Phenolic Compounds

In this paper, 14 compounds were deduced as phenolic compounds in CR-EF herbal pair extract, and were identified based on the MS data (Appendix A). Compounds **4**, **9**, and **14** displayed [M − H]^−^ ion at *m/z* 367, further MS/MS fragmentation of the deprotonated molecular ion was almost the same, a strong fragment ions at *m/z* 161 [caffeic acid – H − H_2_O]^−^ and 135 [caffeic acid – H − CO_2_]^−^or 133 [caffeic acid – H − CO_2_ − H_2_]^−^, in addition, compound **14** showed another characteristic fragment ion at *m/z* 179.0350 [caffeic acid − H]^−^, suggesting 5-substitution [23]. The proposed major fragments were shown in Figure 6. Based on the above arguments, compounds **4**, **9**, and **14** was tentatively identified as 3-*O*-caffeoyl quinic acid methyl ester, 4-*O*-caffeoyl quinic acid methyl ester and 5-*O*-caffeoyl quinic acid methyl ester. The order of elution for caffeoyl quinic acid methyl ester was the same as reported in literature [24]. Alkyl ester of caffeoyl quinic acids are less common in natural products [25]. Thus, we can speculate that the above compounds are artifacts of methylation during acidic methanol extraction. 

Compounds **10**, **21**, and **28** exhibited [M + H]^+^ ion at *m/z* 383 (C_18_H_23_O_9_), which gave the same fragment ion at *m/z* 177 [ferulic acid + H − H_2_O]^+^, 145 [ferulic acid + H − H_2_O − CH_3_OH]^+^ and 117 [ferulic acid + H − H_2_O − CH_3_OH − CO]^+^. According to the elution order, compounds **10**, **21**, and **28** were deduced as 3-*O*-feruloylquinic acid methyl ester, 4-*O*-feruloylquinic acid methyl ester and 5-*O*-feruloylquinic acid methyl ester [23]. 

Compounds **5** and **6** showed [M − H]^−^ ions at *m/z* 371.0986 (C_27_H_31_O_10_) and 339.0718 (C_15_H_15_O_9_), which gave the same MS/MS fragments with those of compound **4** this suggest that compounds **5** and **6** have a caffeate moiety. Compound **5** showed another characteristic fragment ion at *m/z* 339.0732 [M – H − CH_3_OH]^−^, suggesting it has a terminal methoxyl group. Thus, compounds **5** and **6** were tentatively deduced as caffeoyl-6-*O*-d-gluconic acid methylester and caffeoyl-6-*O*-d-glucono-γ-lactone, which were isolated from the nearly ripe fruits of *Evodia rutaecarpa* (Juss.) Benth.[26].

Compound **19** displayed an [M − H]^−^ ions at *m/z* 193.0510 (C_10_H_9_O_4_), further afforded similar fragment ions with compound **4** at 160.8414 [M – H − OCH_3_ − H_2_]^−^, 134.0362 [M – H − CH_3_CO]^−^, 133.0283 [M – H − CH_3_COOH]^−^ in the MS/MS mode, indicating the possible presence of a caffeate moiety. From the literature [27], it can be deduced that caffeic acid derivative commonly exist in EF, so compound **19** was tentatively characterized as methyl caffeate.

Compound **8** exhibited a quasi-molecular ion peak [M − H]^−^ at 355.1029, and its MS/MS spectra yielded a base peak 193.0505 by loss of a glucosyl moiety, which produced the similar fragment ions with compound **19** at 161.0260 [Caffeate – H_2_O]^−^, 134.0349 [M – H – glc – CH_3_COO]^−^. Thus, compound **8** could be tentatively established as methyl 4-*O*-β-d-glucosyl caffeate [28].

Compound **2**, showed an [M − H]^−^ ion at *m/z* 211.0612, and its MS/MS spectra showed a base ion at 123.0458 [M – H − CH_3_COOH − CO]^−^ accompanied by a weak ion at 151.8873 [M – H − CH_3_COOH]^−^. The characteristic ion at *m/z* 123 was the same as reported in literature [29], so compound **2** was tentatively identified as danshensu methyl ester, which was reported previously exsiting in CR [13].

Compound **1** presented [M − H]^−^ ion at *m/z* 315.0721, corresponding to the molecular formula of C_13_H_16_O_9_. In addition, the fragment ions at *m/z* 152.0107 and 108.0212 were generated by loss of glucosyl and followed by HCOO from the parent ion. This compound has the same fragmentation pattern with reported gentisoyl glucoside [30]. Thus, compound **1** was tentatively assigned as gentisic acid-5-*O*-β-d-glucoside, which was previously reported from CR [13].

Compound **7** exhibited an [M − H]^−^ at m/z 521.2030, corresponding to the molecular formula of C_26_H_34_O_11_, which yielded fragment ions at *m/z* 359.1508 and 344.1276 due to the loss of a glucosyl group followed by a loss of methyl group, respectively. Thus, compound **7** was tentatively identified as (+)-lariciresinol glucoside by comparion with the reference data, which was previously reported from CR [13].

Compound **33** displayed an [M + H]^+^ at *m/z* 609.1818, which yielded fragment ions at *m/z* 463.1235 and 301.0706 due to the loss of a rhamnosyl group followed by a loss of glucosyl group, respectively. Thus, compound **33** was tentatively identified as diosmin by comparion with the reference data [31].

### 2.3. UHPLC-DAD Fingerprint Analysis

#### Fingerprints Collection and Similarity Calculation

A total of 24 chromatographic peaks were found in all 19 batches of CR-EF herbal pair samples. A peak at *t*_R_ = 56.635 min was named as an S peak (internal standard peak) to calculate relative retention times (RRTs) and relative peak areas (RPAs) of other peaks. According to the RRTs of the chromatographic peak in each sample, 24 peaks labelled with Arabic numerals were designated as common peaks. Furthermore, after the similarity of “cosine” for samples was evaluated between each other, and the reference fingerprint chromatogram was automatically generated through the software of “Similarity Evaluation System Chromatographic Fingerprint of TCM (2012)” based on the chromatograms of the 19 batches of CR-EF herbal pair samples (Figure 7). The chromatograms of reference fingerprint (R) and 19 batches of different sample fingerprints (S2–S20) are shown in Figure 7. The similarities between each of the 19 samples with the reference fingerprint were all above 0.95 (Table 1). By comparing with the reference substances and LC retention behavior, 24 peaks (except peak 7) were identified as caffeoyl-6-*O*-d-gluconic acid methylester (**1**), magnoflorine (**2**), caffeoyl-6-*O*-d-glucono-γ-lactone (**3**), methyl 4-*O*-β-d-glucosyl caffeate (**4**), 4-*O*-caffeoylquinic acid methyl ester (**5**), 3-*O*-feruloylquinic acid methyl ester (**6**), 5-*O*-caffeoyl quinic acid methyl ester (**8**), methyl caffeate (**9**), 4-*O*-feruloylquinic acid methyl ester (**10**), 8-oxoepiberberine (**11**), dehydroevodiamine (**12**), coptisine (**13**), 4-*O*-feruloylquinic acid methyl ester (**14**), epiberberine (**15**), columbamine (**16**), jatrorrhizine (**17**), 5-*O*-feruloylquinic acid methyl ester (**18**), berberine (**19**), palmatine (**20**), evodiamine (**21**), rutaecarpine (**22**), evocarpine (**23**), and 1-methyl-2-[(*Z*)-10-pentadecenyl]-4(1*H*)-quinolone (**24**).

### 2.4. Selection and Confirmation of Marker Constituents

The qualitative indicated that 44 alkaloids, seven limonoids, and 14 phenolic compounds were the major constituents in the CR-EF herbal pair (Appendix A). Thus, alkaloids were the major components both in CR and EF. Among them, berberine, palmatine, coptisine, epiberberine, columbamine and jatrorrhizine, all belonging to protoberberine alkaloids, were the most abundant alkaloids in CR [7]. Berberine, palmatine, coptisine, epiberberine, and jatrorrhizine, were also the main constituents in the blood of rats administered with CR [32]. 

Upon further investigation, berberine was reported to alleviate ischemic arrhythmias [33], mitigate cognitive decline in a triple-transgenic mouse model of Alzheimer’s disease [34], and ameliorate lipopolysaccharide-induced inflammatory response [35]. Palmatine has been proved as an anti-inflammatory drug for clinical treatment of gastrointestinal infections, surgical infections, and gynecological inflammation [36]. Coptisine decreased cholesterol biosynthesis by down-regulating mRNA and expression of HMGCR [37], and jatrorrhizine promoted the conversion of cholesterol into bile acids and then reduce the accumulation of cholesterol in blood [38]. Columbamine showed excellent blood lipid-lowering activities and could reverse the body weight gain of hamsters induced by HFHC (high-fat, high-carbohydrate) diet [39]. In addition, columbamine presented as being non-cytotoxic to human dermal fibroblasts but exhibited a cytotoxic effect against B16-F10 melanoma cells and HepG2 cells [40,41]. Columbamine suppression was also confirmed on subcutaneous tumor growth in nude mice tumor model produced with SMMC7721 cells [42].

Evodiamine, dehydroevodiamine, and rutaecarpine, belonging to indoloquinazoline alkaloids having a five unbroken ring skeleton, were the major bioactive alkaloids in EF [43], and had been found antitumor effects in several cancer cell lines, such as hepatocellular carcinoma cell lines [44], renal carcinoma cell lines [45], human bladder cancer cell line [46], multiple myeloma cells [47], human lung cancer cell lines [48], etc., as well as protective effects on the cardiovascular system [49,50,51], vasodilatory effects on endothelium [52,53], anti-inflammatory effects [54,55,56], and neuroprotective effects [51].

From the above investigation, alkaloids were regarded as important activity biomarkers in the CR-EF herbal pair. Consequently, nine high-content or strong bioactivity compounds, including coptisine, epiberberine, columbamine, jatrorrhizine, berberine, palmatine, dehydroevodiamine, evodiamine, and rutaecarpine, were selected as markers for quantification (Figure 8).

In addition to the high-resolution mass spectrometry data, these nine markers were confirmed by comparison with the reference standards, and their retention times and UV spectra matched very well. The peak purities could be evaluated by using UV absorption at each sampling point in one peak by DAD. As a result, similarity factors of nine peaks were above 0.995 in the 200–400 nm wavelength range, which indicated that all the marker peaks had high purities.

### 2.5. UHPLC-DAD Quantitative Analysis

Quantitative determinations of the above selected nine compounds were performed by UHPLC-DAD.

#### 2.5.1. Linearity and Limit of Quantification

External standard calibrations were established at six data points covering the concentration range of each compound according to the level estimated in the plant samples. The calibration curves were constructed by plotting the peak areas against the concentration of each analyte. As shown in Table 2, all nine analytes showed good linearity (R^2^ > 0.999) within the test range and verifited by applying the ANOVA (*p* < 0.001) performed at the 99% confidence level. The lower limit of detection (LLOD) and lower limit of quantification (LLOQ) for each analyte under the present chromatographic conditions were determined at a signal-to-noise (*S*/*N*) ratio of about 3 and 10, respectively.

#### 2.5.2. Precision, Repeatability, and Stability

Precision, repeatability, and stability of the method were validated for each analyte. Intra-day and inter-day variations were used to determine the measurement precision of the developed method. For the intra-day variability test, a sample solution, prepared as described in Section 3.4, was analysed for six replicates within one day, while for the inter-day variability test, the sample was tested in duplicates for three consecutive days. To confirm the repeatability, six replicates of the same sample were extracted and analysed as described in Section 3.4. A sample solution was examined at 0, 2, 4, 8, 12, and 36 h to evaluate sample stability. Relative standard deviations (RSDs) of intra-day and inter-day precision, repeatability, and stability were all ≤ 2.0% (Table 3).

#### 2.5.3. Recovery

The accuracy of quantification method was validated by a standard spiking test. The proposed method was applied to the samples spiked with the mixed standard solution at 100% concentration level. Six replicates experiments at each level were performed. The ratio of detected and added amounts was used to calculate the recovery. Average recoveries of the nine analytes ranged from 92.94% to 101.47% with all of the RSDs ≤ 2.0%, suggesting that the method was accurate (Table 3).

#### 2.5.4. Sample Analysis

The quantity of bioactive compounds in the CR-EF herbal pair is very important for its therapeutic effects. The newly developed and validated method was applied to 19 batches of samples and the quantification results of the nine analytes are summarized in Appendix A. It can be seen that the applied chromatographic conditions brought a good separation for these nine compounds, and they could be detected in all samples. Six alkaloids from CR including coptisine, epiberberine, columbamine, jatrorrhizine, berberine, and palmatine, which were regarded as the important biomarkers for several activities, were found to be the major ingredients, counting for 11.10–18.52, 6.78–11.85, 3.91–9.16, 2.71–4.76, 33.41–44.87, and 7.89–13.30 mg·g^–1^, respectively. The contents of dehydroevodiamine, evodiamine, and rutaecarpine, three major marker alkaloids in EF, ranged from 1.50–2.10, 0.15–3.67, and 0.13–1.73 mg·g^–1^, respectively. The reasons responsible for the variation may include different manufacturing processes and harvest times of the compositional herbal materials. Therefore, it is significant to determine as many bioactive components as possible for quality evaluation of these preparations containing the CR-EF herbal pair.

## 3. Materials and Methods

### 3.1. Crude Drug Materials

Coptidis Rhizoma (CR1–CR20) were collected from different areas of Hubei and Sichuan provinces and Chongqing city of China. Euodiae Fructus (EF1–EF20) was collected from different areas of Hunan and Jiangxi provinces of China. All samples were authenticated by Professor Hui Cao (School of Pharmacy, Jinan University, Guangzhou, China). These samples have been deposited in the Beijing Increase Innovative Drug Co. Ltd., Beijing, China. The 20 batches of Coptidis Rhizoma and Euodiae Fructus herbal pair samples (S1–S20) are summarized in Appendix A.

### 3.2. Chemical and Reagents

Reference standards of berberine hydrochloride (lot 110713-201814, purity > 86.8%), palmatine chloride (lot 110732-201611, purity > 86.8%), evodiamine (lot 110802-201409, purity > 99.4%), rutaecarpine (lot 110801-201608, purity > 99.5%), and jatrorrhizine chloride (lot 110733-201609, purity > 89.5%) were acquired from National Institues for Food and Drug Control (Beijing, China); coptisine chloride (lot 16061705, purity > 99.5%) were obtained from Beijing Beina Chuanglian Biotechnology Research Insitute (Beijing, China). Epiberberine (lot 18101203, purity > 98%) and columbamine (lot 18011702, purity > 91%) were purchased from Chengdu Pufei De Biotech Co., Ltd. (Chengdu, China). Dehydroevodiamine (lot 5451, purity > 99.8%) and magnoflorine (lot 3536, purity > 98%) were obtained from Shanghai Standard Technology Co., Ltd. (Shanghai, China).

LC-MS grade acetonitrile and formic acid (Thermo Fisher Scientific, NY, USA), trifluoroacetic acid (TFA) (Shanghai Titian scientific, Shanghai, China), and ultra-pure water (Watsons Group Co., Ltd. Hong Kong, China) were used in the mobile phase. Methanol (LC-grade) and hydrochloric acid (HCl, analytical reagent) for sample preparation were obtained from Tianjin Saifurui Co., Ltd. (Tianjin, China) and Beijing Yili Fine Chemicals Co., Ltd. (Beijing, China), respectively.

### 3.3. Chromatographic and Mass Spectrometric Conditions

Qualitative analysis was performed on an Agilent 1290 Infinity II UHPLC system coupled to an G6530C Q-TOF-MS mass spectrometer, equipped with a quaternary pump, a diode-array detector (DAD), a column oven, an autosampler, and an ESI source (Agilent Technologies, Inc., Palo Alto, CA, USA), and data analysis was performed on Agilent Qualitative Navigator (B.08.00) software and Qualitative Workflows (B.08.00) software. Quantitative determination was performed on an Agilent 1290 Infinity II UHPLC system and a Waters ACQUITY UPLC H-Class PLUS system (Waters Corporation, Milford, MA, USA), equipped with a quaternary pump, a column oven, an autosampler, and photodiode array detectors (PDA).

All qualitative and quantitative separation was successfully performed on a Welch^TM^ Ultimate UHPLC AQ-C_18_ column (2.1 × 100 mm, 1.8 µm). The mobile phases for the LC-DAD fingerprint and quantitative method were composed of (A) acetonitrile and (B) 0.01% TFA aqueous solution, and for the LC-MS qualitative method were consisted of (A) acetonitrile and (B) 0.05% formic acid aqueous solution. The optimized gradient elution was described as follows: 0–25 min, 5–11% A; 25–35 min, 11% A; 35–50 min, 11–13% A; 50–60 min, 13–20% A; 60–70 min, 20–55% A; 70–75 min, 55–90% A; 75–80 min, 90% A. The flow rate was 0.4 mL·min^–1^, the inject volume was 1 µL, and the column temperature was set at 30 °C. The detection wavelength was set at 300 nm.

During qualitative analysis, the mass spectrometry detection was acquired in both positive and negative ion modes. The drying temperature was 350 °C, the drying gas flow rate was 10 L·min^–1^, the pressure of nebulizer gas was 35 psi, the sheath gas temperature was 350 °C, the sheath gas flow rate was 12 L·min^–1^, the capillary voltage was 4000 V (positive mode) and 3500 V (negative mode). MS mode was selected for primary mass spectrometry with a quality scanning range of *m/z* 100–1300. Auto-MS/MS mode was used for the secondary mass spectrum, and the collision voltage was 10, 20, and 30 V. LC-MS data were collected by Agilent MassHunter (B.08.00) software.

### 3.4. Preparation of Sample and Standard Solutions

#### 3.4.1. Preparation of Sample solutions

Twenty batches of each crude drug were refined and processed to obtain Coptidis Rhizoma and Euodiae Fructus decoction pieces.

Coptidis Rhizoma and Euodiae Fructus (6:1) herbal pair (CR-EF herbal pair) powder: Coptidis Rhizoma and Euodiae Fructus decoction pieces were ground into powders with a particle size of 50 mesh before use. Powders of Coptidis Rhizoma (18 g) and Euodiae Fructus (3 g) were mixed according to batches, as shown on Appendix A.

An amount of 0.7 g CR-EF herbal pair powder was accurately weighed and soaked in 100 mL of MeOH-HCl solution (100:1; *v*/*v*) for 1 h, and then ultrasonic-extracted for 30 min. The weight loss in the ultrasonic-extraction procedure was compensated and the extracted solution was centrifuged at 13,000 r·min^–1^ for 10 min, then the supernatant was removed to analysis.

#### 3.4.2. Preparation of Standard Solutions

The reference standards of dehydroevodiamine, coptisine chloride, epiberberine, columbamine, jatrorrhizine, berberine hydrochloride, palmatine chloride, evodiamine, and rutaecarpine were dissolved with MeOH and stocked at 4 °C, respectively.

The reference compound stock solutions of coptisine chloride, epiberberine, berberine hydrochloride, and palmatine chloride were mixed to obtain the mixed reference standard solution A and B. The final concentration of A were 234.36, 168.75, 590.85, 197.04 μg·mL^–1^, respectively. The final concentration of B were 93.74, 67.50, 236.34, 78.82 μg·mL^–1^, respectively.

The reference compound stock solutions of evodiamine and rutaecarpine were mixed to obtain the mixed reference standard solution C and D. The final concentration of C were 42.00 and 21.12 μg·mL^–1^. The final concentrations of D were 4.20 and 2.11 μg·mL^–1^.

The reference compound stock solutions of dehydroevodiamine and jatrorrhizine were mixed to obtain the mixed reference standard solution E and F. The final concentration of E were 42.96 and 149.02 μg·mL^–1^. The final concentrations of F were 4.30 and 56.61 μg·mL^–1^.

The reference compound stock solutions of columbamine was then diluted with MeOH to obtain the final concentrations of 71.11 and 17.78 μg·mL^–1^, which were solution G and H.

### 3.5. UHPLC-DAD Method Validation

The linearity, precisions, repeatability, stability, as well as the accuracy were evaluated to validate the proposed UHPLC-DAD method.

The lower limit of detection (LLOD) and lower limit of quantification (LLOQ) for nine alkaloids were estimated at signal-to-noise ratios (*S*/*N*) of 3 and 10, respectively, by injecting a series of dilute solutions with known concentration.

Solutions A–H were injected to detection with appropriate volume, and the calibration curves were established by determining the peak areas against the compound amount (µg) of each analyte.

The intra- and interday precisions were validated with mixed solutions under the optimized conditions six times in the same day and once a day for three consecutive days, respectively. The repeatability was performed by analyzing six independently prepared sample solutions. The stability was carried out with one sample solution intervals of 0 h, 2 h, 4 h, 8 h, 12 h, and 36 h.

For the recovery test, accurate amounts of the nine alkaloids were spiked into 0.35 g of CR-EF herbal pair powder (S1) and then extracted and analyzed.

### 3.6. Data Analysis

Similarity analysis (SA) was evaluated by professional software named “Similarity Evaluation System for Chromatographic Fingerprint of Traditional Chinese Medicine (Version 2012)” designated by the Chinese Pharmacopoeia Committee.

## 4. Conclusions

In this study, a systematic method for quality control of the CR-EF herbal pair was established, which can realize both qualitative and quantitative analysis. In UHPLC-QTOF-MS quality analysis, 65 compounds, mainly including alkaloids, phenolic compounds, and limonoids, in the CR-EF herbal pair system were identified or tentatively characterized. According to the qualitative analysis results, an optimized UHPLC-DAD method, which showed good precision, repeatability, and recovery, was then established for the chemical fingerprint analysis of the CR-EF herbal pair and the quantitative determination of nine major alkaloids. At the same time, the validated method was used to evaluate the quality of CR-EF herbal pair commercial preparations. This method facilitates the improvement of the quality control of this formula. In a word, the present study provided a stable and reliable method for quality evaluation of the CR-EF herbal pair, which could provide a basis for pharmacological research, clinical applications, and guiding new drug development of effective CR-EF herbal pair compositions.

## Figures and Tables

**Figure 1 molecules-25-04782-f001:**
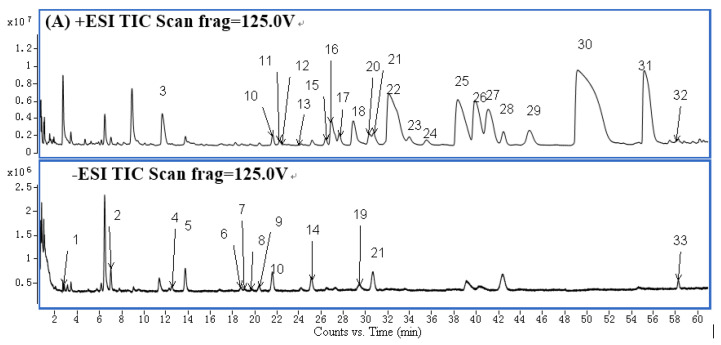
TIC of the CR-EF herbal pair extract of 0–60 min (**A**) and 60–80 min (**B**). Peaks have been numbered according to the Appendix A.

**Figure 2 molecules-25-04782-f002:**
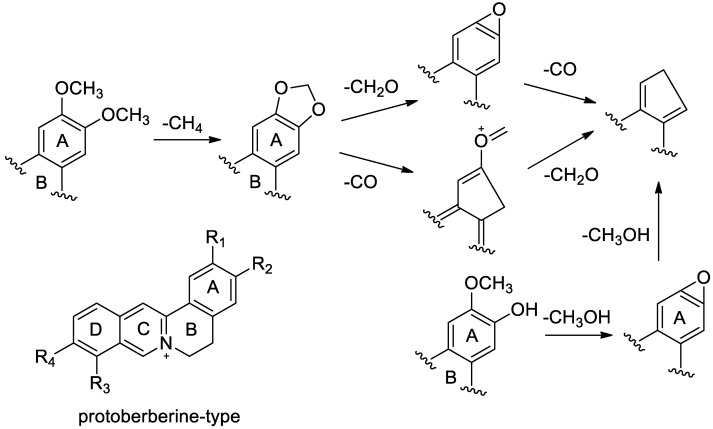
General MS/MS fragmentation pathways of protoberberine-type alkaloids.

**Figure 3 molecules-25-04782-f003:**
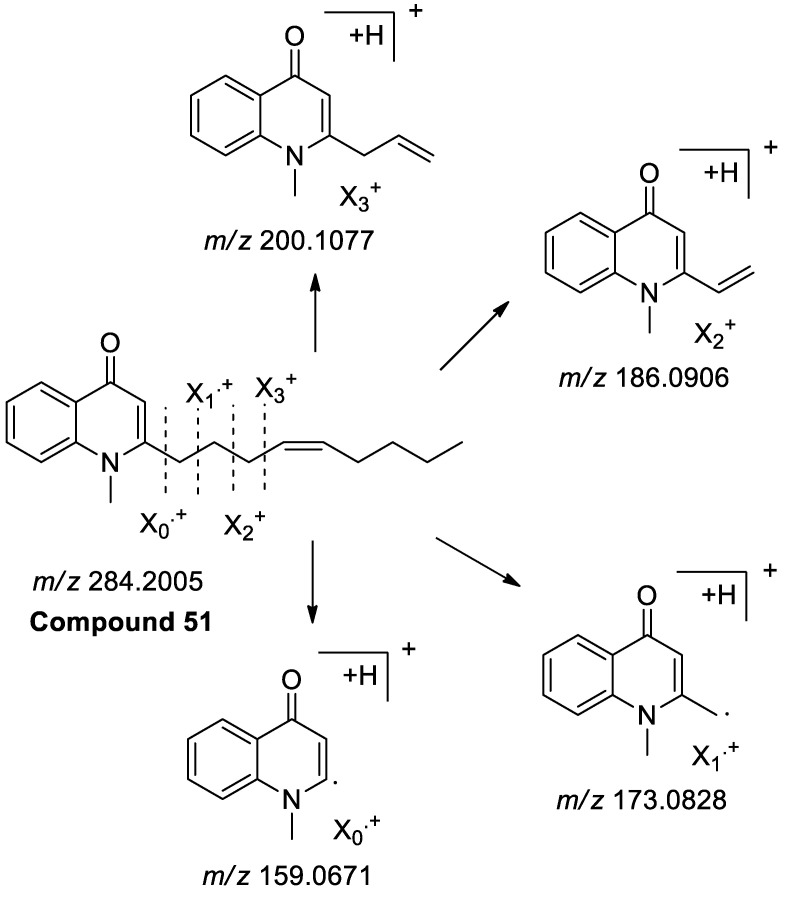
MS/MS proposed fragmentation pathways of 1-methyl-2-[(*Z*)-4-nonenyl]-4(1*H*)-quinolone.

**Figure 4 molecules-25-04782-f004:**
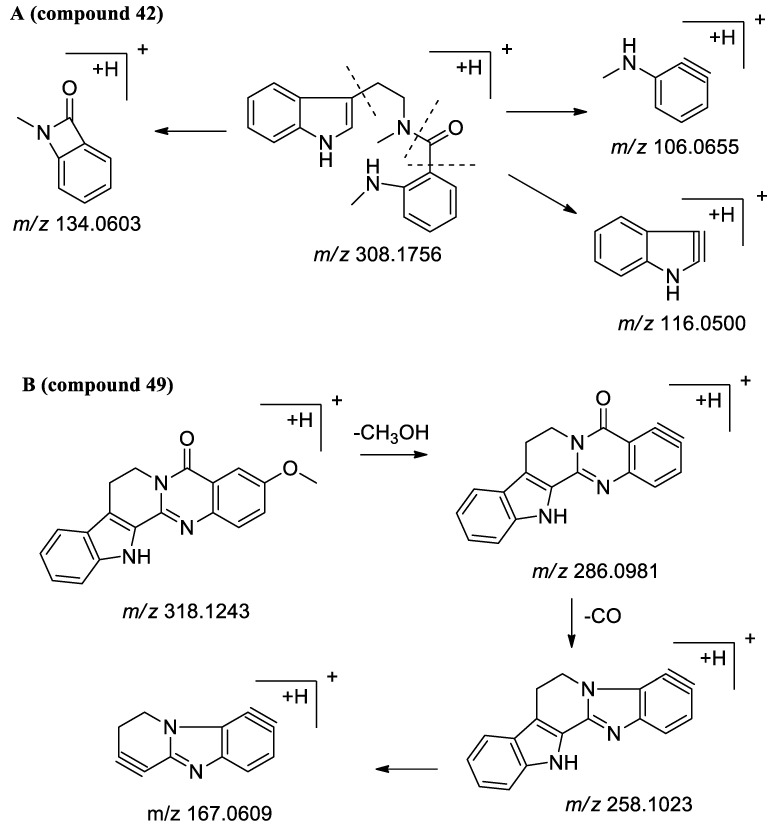
MS/MS proposed fragmentation pathways of evodiamide (**A**) and hortiacine (**B**).

**Figure 5 molecules-25-04782-f005:**
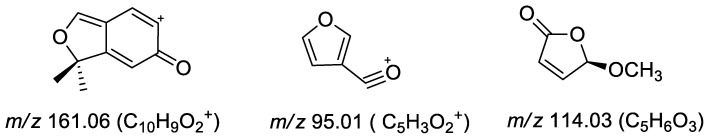
Proposed characteristic fragments’ structures of limonoids.

**Figure 6 molecules-25-04782-f006:**
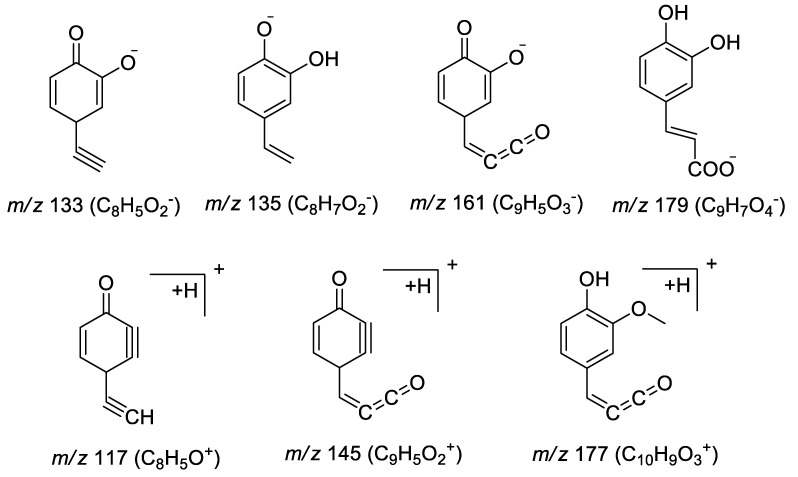
Proposed major fragments of phenolic compounds.

**Figure 7 molecules-25-04782-f007:**
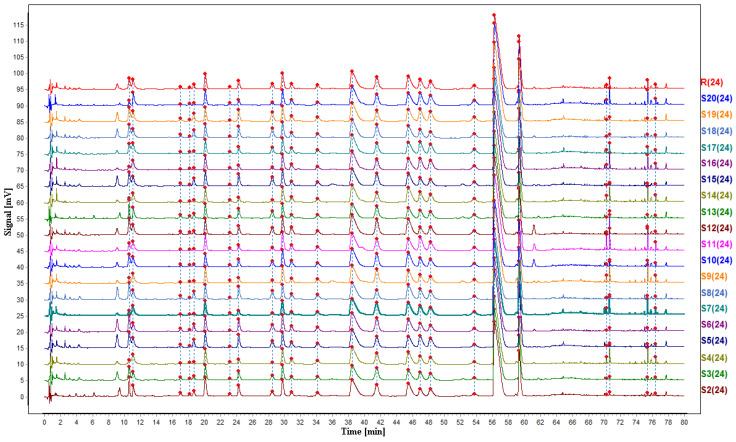
UHPLC fingerprints of different CR-EF herbal pair samples and reference fingerprint.

**Figure 8 molecules-25-04782-f008:**
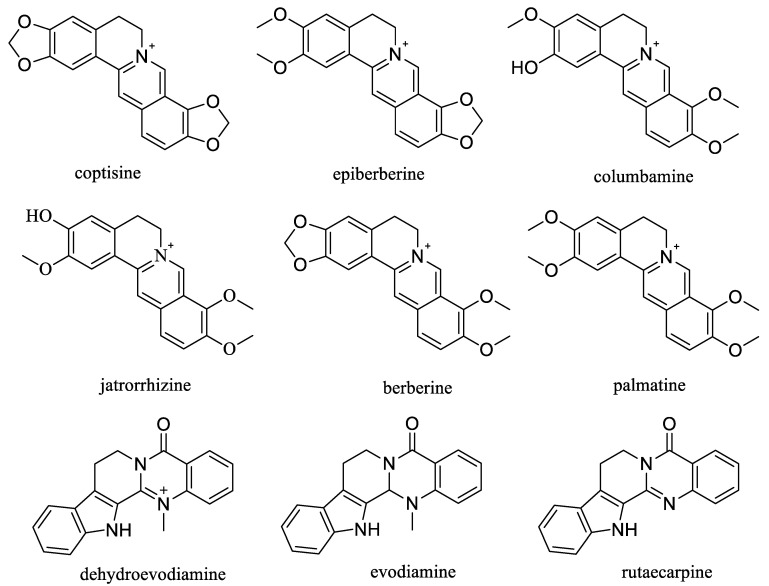
Chemical structures of nine markers for quantification.

**Table 1 molecules-25-04782-t001:** Similarity values of tested samples.

Sample No.	Similarity	Sample No.	Similarity
S2	0.998	S12	0.996
S3	0.998	S13	0.998
S4	0.999	S14	0.998
S5	0.999	S15	0.998
S6	0.997	S16	0.998
S7	0.999	S17	0.999
S8	0.997	S18	0.997
S9	0.998	S19	0.999
S10	0.995	S20	0.998
S11	0.994		

**Table 2 molecules-25-04782-t002:** The regression equations of nine analytes.

Analytes	Linear Range µg·mL^–1^	Calibration Curve	R^2^	ANOVA *p* Value	LLOD µg·mL^–1^	LLOQ µg·mL^–1^
Coptisine	9.37–468.72	Y = 2225.6x − 5608.1	0.9998	0.0000	0.63	2.10
Epiberberine	6.75–337.50	Y = 2120.9x − 3536	0.9998	0.0000	0.81	2.16
Columbamine	3.56–142.22	Y = 2.3547x − 0.6865	0.9997	0.0000	0.81	2.71
Jatrorrhizine	5.96–298.04	Y = 3.3457x − 7.1431	0.9999	0.0000	0.62	1.85
Berberine	23.63–1181.70	Y = 3358x − 12774	0.9998	0.0000	0.53	1.60
Palmatine	7.88–394.08	Y = 3696.4x − 3244	0.9998	0.0000	0.31	0.92
Dehydroevodiamine	4.30–85.92	Y = 3.026x − 1.9044	0.9999	0.0000	0.43	1.29
Evodiamine	0.42–42.00	Y = 0.8719x − 0.1126	0.9998	0.0000	0.12	0.35
Rutaecarpine	0.21–21.12	Y = 5.5394x − 0.6834	0.9996	0.0000	0.06	0.17

**Table 3 molecules-25-04782-t003:** Precision, stability, repeatability and accuracy of nine analytes.

Analytes	Precision		Repeatability		Stability	Recovery	
	Intra-Day RSD% (*n* = 6)	Inter-Day RSD% (*n* = 3)	Mean Concentration (mg·g^–1^)	RSD% (*n* = 6)	RSD% (*n* = 6)	Average Recovery (%)	RSD%
Coptisine	0.4	0.8	14.89	0.9	1.7	95.83	1.8
Epiberberine	0.5	0.6	10.92	1.6	1.7	101.47	2.0
Columbamine	14	1.4	6.56	0.8	0.8	93.70	0.6
Jatrorrhizine	0.6	1.6	3.54	0.8	1.5	93.28	0.8
Berberine	04	0.7	41.54	1.1	1.3	95.19	1.7
Palmatine	0.5	0.7	12.47	2.0	1.2	97.72	1.8
Dehydroevodiamine	1.9	1.7	1.54	1.6	1.6	97.05	1.5
Evodiamine	0.9	1.6	0.94	1.3	1.6	92.94	1.9
Rutaecarpine	2.0	1.3	0.39	1.4	1.4	99.94	1.3

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
