# Peer review of "Simultaneous Qualitative and Quantitative Evaluation of the Coptidis Rhizoma and Euodiae Fructus Herbal Pair by Using UHPLC-ESI-QTOF-MS and UHPLC-DAD"

_molecules, 2020, doi:10.3390/molecules25204782_

Round 1

Reviewer 1 Report

Reviewer report on manuscript Molecules-941182

The submitted paper provides data regarding to the simultaneous qualitative and quantitative
evaluation of Coptidis Rhizoma and Euodiae Fructus herbal pair by using UHPLC-ESI-QTOF-MS and
UHPLC-DAD.

In general terms the manuscript is well-structured, figured and referenced. I have no comments on
the experimental part of this work. The topic of the proposed research fits to the aims and scope of
the present Journal. The optimization of method parameters has been carefully investigated.

However, there are some comment in the text that must be addressed prior to its publication.

Comments

1) Section 2.1. For which analytical method (UHPLC-DAD or UHPLC-ESI-QTOF-MS) the
optimization step referred? The section tile should be revised.
2) Table 3: the RSD values should be expressed with one decimal point.
3) Did the authors employ internal standard on the quantitation of samples?

Author Response

1. Setion 2.1. for which analytical method (UHPLC-DAD or UHPLC-ESI-QTOF-MS) the optimization step referred? The section tile should be revised.

Reply: The section title is revised to “Selection of extraction and analytical conditions”

2. Table 3: the RSD values should be expressed with one decimal point.

Reply: All the RSD values of Table 3 are corrected to one decimal point.

3. Did the authors employ internal standard on the quantitation of samples?

Reply: We did not employ internal standard on the quantitation of samples.

Reviewer 2 Report

I find the presented manuscript interesting, but I have some minor concerns:

  • the language must be corrected
  • on the figure 2 there is proposed an oxonium ion, in such case the analysed ion would have z=2 and m/z equal to half of the monoisotopic mass of the ion - was this fact taken under concideration during the analysis?
  • line 339 - similarity factor 0.995, not 995
  • line 349 - R2 is a coefficient of determination and does not necesarly describe the linerarity, there are other linearity tests which should be used, like Mandel's fitting test.

I recommend acceptance of the manuscript after minor revision.

Author Response

1. on the figure 2 there is proposed an oxonium ion, in such case the analysed ion would have z=2 and m/z equal to half of the monoisotopic mass of the ion - was this fact taken under concideration during the analysis?

Repyl: We did not find the mass fragment of half of the monoisotopic mass of the ion. We supposed that the oxonium ion was produced by losing the neutral CO molecule and the migration of one positive ion from N to O. So the oxonium ion fragment has z=1.

2. line 339 - similarity factor 0.995, not 995

Reply: This experiment was performed on an Agilent 1290 InfinityⅡUHPLC system with LC OpenLab CDS 2 software. After consulting with the engineering, the engineer indicated that the full score of purity of the instrument was 1000, so the presentation form was 995. If the full score was converted to 1, it was 0.995。So 995 is amended for “0.995”.

3. line 349 - R2is a coefficient of determination and does not necessary describe the linerarity, there are other linearity tests which should be used, like Mandel's fitting test.

Reply: The regression equation was verified by applying the ANOVA (SPSS 16.0), and the p values of the nine compounds were all less than 0.001, indicating a significant linear relationship. And the p values are added to the manuscript.

Reviewer 3 Report

The article “Simultaneous qualitative and quantitative evaluation of Coptidis Rhizoma and Euodiae Fructus herbal pair by using UHPLC-ESI-QTOF-MS and UHPLC-DAD” aimed to established a systematic method for chemical profiling and quantification analysis of the major constituents in CR-EF herbal pair. The article is interesting, however I recommend some minor changes:

  1. Title: The Authors propose UHPLC-DAD as cheaper and widely available method for qualitative and quantitative evaluation of Coptidis Rhizoma and Euodiae Fructus herbal pair. As I well understood LC-MS was used only to perform identification of the most abundant compounds. Please correct.
  2. What is the cut-off of “similarity” to assign the formulation as “sufficient quality” and “insufficient quality”? Have you checked the fingerprint of pair made with different ratio of rhizome/fructus?
  3. Different abbreviation are used for the same technique used i.e. UHPLC-DAD-TOF-MS/MS, UHPLC-ESI-QTOF-MS. Please standardize
  4. Line 37. Please add the reference
  5. Line 58. Are they mean values? It would be better to provide the range or at least mean and standard deviation
  6. Line 96. TFA is not MS-friendly mobile phase modifier. Moreover, it is were hard to remove the acid from the LC-MS after the analysis. Thus, many laboratories select methods without TFA. Similarly- phosphoric acid is not used in MS analysis.
  7. Line 387-390. Were the harvest time similar for all samples? Was it harvested as recommended by Pharmacopoeia?

Author Response

1. Title: The Authors propose UHPLC-DAD as cheaper and widely available method for qualitative and quantitative evaluation of Coptidis Rhizoma and Euodiae Fructus herbal pair. As I well understood LC-MS was used only to perform identification of the most abundant compounds. Please correct.

Reply: ①UHPLC-DAD cannot be used for qualitative analysis.  ②The sensitivity of LC-QTOF-MS is higher than that of UHPLC-DAD, and mainly used for the identification of unknown compounds, and also can analyze the minor components less than 1ppm. ③ In this paper, qualitative analysis by using UHPLC-QTOF-MS and the quantitative analysis by using UHPLC-DAD. So we think that there is no need to correct the title.

2. What is the cut-off of “similarity” to assign the formulation as “sufficient quality” and “insufficient quality”? Have you checked the fingerprint of pair made with different ratio of rhizome/fructus?

Reply:

① Generally, it is considered to be of good quality (sufficient quality) that the similarity is above 0.90.

②This fingerprint analysis method can be used different ratio of CR-EF herbal pair, but the reference fingerprint from this paper cannot be used as the reference to evaluate the other ratio of herbal pair.

3. Different abbreviation are used for the same technique used i.e. UHPLC-DAD-TOF-MS/MS, UHPLC-ESI-QTOF-MS. Please standardize

Reply:

“UHPLC-DAD-QTOF-MS/MS” is revised to “UHPLC-ESI-QTOF-MS”, and all “UHPLC-ESI-QTOF-MS/MS” is also revised to “UHPLC-ESI-QTOF-MS”.

4. Line 37. Please add the reference

Reply: added the reference [1].

5. Line 58. Are they mean values? It would be better to provide the range or at least mean and standard deviation

Reply: They are mean values and added the standard deviation. And amended as follows.

“It is reported that the quantities of seven alkaloids in ethanol extract [4] of Zuojin pill are epiberberine 32.65 ± 0.43 mg·g–1, jatrorrhizine 13.59 ± 0.15 mg·g–1, coptisine 110.64 ± 1.15 mg·g–1, palmatine 61.20 ± 0.79 mg·g–1, berberine 153.0 ± 1.94 mg·g–1, evodiamine 1.89 ± 0.026 mg·g–1, and rutaecarpine 1.47 ± 0.050 mg·g–1, while the quantities of six alkaloids in the water extract [7] of Zuojin pill are columbamine 2.81 ± 0.08 mg·g–1, epiberberine 3.90 ± 0.07 mg·g–1, jatrorrhizine 2.70 ± 0.04 mg·g–1, coptisine 6.85 ± 0.08 mg·g–1, palmatine 11.02 ± 0.11 mg·g–1, and berberine 35.86 ± 0.72 mg·g–1.”

6. Line 96. TFA is not MS-friendly mobile phase modifier. Moreover, it is were hard to remove the acid from the LC-MS after the analysis. Thus, many laboratories select methods without TFA. Similarly- phosphoric acid is not used in MS analysis.

Reply: This is clerical error, and corrected as follows.

“It was found that the suitable mobile phase is the mixture of acetonitrile and 0.01% (v/v) TFA aqueous solution, which make it possible to separate 24 compounds simultaneously. Considering TFA unsuitable for MS system, so 0.05% formic acid was selected for MS analysis.”

7. Line 387-390. Were the harvest time similar for all samples? Was it harvested as recommended by Pharmacopoeia?

Reply: All crude drug materials were harvested as recommended by Pharmacopoeia.

Round 2

Reviewer 1 Report

The authors addressed properly my previous comments